# Evaluating the state-of-the-art in mapping research spaces: A Brazilian case study

**Francisco Galuppo Azevedo, Fabricio Murai**⊙*

Department of Computer Science, Universidade Federal de Minas Gerais, Belo Horizonte, MG, Brazil

* murai@dcc.ufmg.br

## Abstract

Scientific knowledge cannot be seen as a set of isolated fields, but as a highly connected network. Understanding how research areas are connected is of paramount importance for adequately allocating funding and human resources (e.g., assembling teams to tackle multidisciplinary problems). The relationship between disciplines can be drawn from data on the trajectory of individual scientists, as researchers often make contributions in a small set of interrelated areas. Two recent works propose methods for creating research maps from scientists' publication records: by using a frequentist approach to create a transition probability matrix; and by learning embeddings (vector representations). Surprisingly, these models were evaluated on different datasets and have never been compared in the literature. In this work, we compare both models in a systematic way, using a large dataset of publication records from Brazilian researchers. We evaluate these models' ability to predict whether a given entity (scientist, institution or region) will enter a new field w.r.t. the area under the ROC curve. Moreover, we analyze how sensitive each method is to the number of publications and the number of fields associated to one entity. Last, we conduct a case study to showcase how these models can be used to characterize science dynamics in the context of Brazil.

## Introduction

It is well known that the amount of resources devoted by a country (e.g., funding, scientists) to certain research fields is highly correlated with technological innovation in the corresponding industries [1–3]. For instance, Department of Defense/NASA expenditures are linked to patents in the aerospace industry. However, scientific knowledge cannot be seen as a set of isolated fields, but as a highly connected network, thus rendering the task of resource allocation extremely challenging [4]. Solutions to important scientific problems often require the interaction between multiple expertises, underlining the importance of interdisciplinary research groups. This relationship between disciplines can also be drawn from the trajectory of individual scientists, as researchers often make contributions in a small set of interrelated areas [5]. Mapping the links between disciplines and hence, the structure of scientific knowledge, is paramount to understand the dynamics of science and its impact on the development of countries [6].

**Data Availability Statement:** The data underlying this study are available on the Zenodo database (https://doi.org/10.5281/zenodo.4288583).

**Funding:** The author(s) received no specific funding for this work.

**Competing interests:** The authors have declared that no competing interests exist.

Recent approaches based on publication records enabled the unveiling of links between disciplines entirely based on data, being less susceptible to subjectivity biases. Guevara *et al*. [5] proposed a Frequentist model, which considers the fraction of researchers in a field that also publish in other fields. In constrast, the work of Chinazzi *et al*. [7] is based on a machine learning method—the StarSpace algorithm [8]—to create embeddings (vector representations) for each reseach field, henceforth referred as Embedding model. Jaffe *et al*. [6] proposed a model that represents a research field as an array where each coordinate is the fraction of the research output of a country associated with that field.

These models allow us to predict a scientist's potential of making contributions in a new research area. From another perspective, they allow us to foresee possible career paths for them. Moreover, we can group researchers based on institution affiliation or on geographic regions and make predictions regarding the scientific areas they will enter in the near future.

Surprisingly, none of these models have been compared in the literature. Each one was tested on a different dataset. In this context, we hereby make **the following contributions**:

- We build a new dataset containing about 2.3 million publication records from almost 175 thousand Brazilian researcher profiles collected from Lattes—the official platform in Brazil for storing data about researchers—and make it available to the research community at https://doi.org/10.5281/zenodo.4288583.

- We compare the Frequentist and the Embedding models in a systematic way, using the aforementioned dataset. We evaluate these models on the ability of predicting that a given entity (scientist, institution or region) will enter a new field. As in previous works, we consider the area under the ROC curve (AUROC) as the performance measure.

- We analyze how sensitive each method is to the number of publications and the number of fields associated to one entity.

- We conduct a case study to showcase how these models can be used to characterize science dynamics in the context of Brazil. We include specific examples of entities which illustrate the strengths and shortcomings of each model.

The rest of this paper is organized as follows. First, we review the body of work related to ours. We then describe the steps involved in the acquisition and preparation of the data used in this paper. In the following two sections, we describe the two research space models and the experimental setup we use to evaluate them. We then present and discuss the results of the evaluation. Next, we present the Brazilian case study. Finally, we discuss our conclusions in the last section.

## Related work

The interest in mapping the relationship between fields of knowledge dates back at least to the XIV century, when Ramon Llull created the first known science tree. The ideas of categorizing knowledge directly have been very influential until today, and have inspired the creation of the University of California San Diego (UCSD) Science Map and Classification System, used by some universities to categorize the production of their scholars [9].

### Methods for crafting research maps

Currently, there are two main approaches for creating such maps. The first is through citations and the second is through career paths, i.e., the trajectory of scholars. The citation-based approaches can be subdivided into: those based on co-citation, which connect areas of papers that co-appear in the reference list of another paper; those based on direct citations, which

connect areas of papers that cite each other directly; and those based on bibliographic coupling, which connect areas of papers that cite the same papers. On the other hand, approaches based on career paths leverage the fact that scholars work in multiple fields during their careers. Therefore, we can infer relationships between research areas from their trajectories. Among the citation-based maps, the UCSD Science Map and Classification System, based on bibliographic coupling, is one of the most marked examples. However, since a career path-based map proposed by Guevara *et al.* [5] was shown to outperform the previous method when predicting the fields that individuals and organizations will enter in the future, we focus on methods that build research spaces from the trajectories of individual researchers.

Guevara *et al.* propose a probabilistic model based on career paths to map the relationship between research fields [5]. The model takes as input a set of publication records along with metadata on the research fields associated with venue. Based on the joint probabilities of publishing in a pair of fields, the model estimates the probability that a scientist joins a new field. The authors collect Google Scholar profiles and use the Scopus classification system to categorize venues. They compare the proposed model with the UCSD Science Map when predicting which new fields scientists, institutions and countries will join, showing that the proposed method significantly outperforms the baseline except when predicting transitions for countries. This model, henceforth referred as the Frequentist model, is explained in detail in Section Models, as it is one of the methods we consider in our evaluation.

The second model is based on vector representations, also known as embeddings. Embeddings have become very popular since the word2vec method (as in "word to vector") was proposed to create dense representations for words. Essentially, they allow us to map elements of a set (typically represented as sparse, one hot encodings) to vectors in a space with dimension much lower than the set size, where either distance or cosine similarity can be used to measure the (dis)similarity between items. Chinazzi *et al.* proposes a method that builds on the StarSpace model to create embeddings for research fields. The authors evaluate their method using the articles published in the American Physical Society between 1986 and 2009 using the Physics and Astronomy Classification Scheme (PACS) codes reported in each publication. They consider the tasks of predicting specialization of urban areas, but do not compare with any baseline. Instead, the authors show correlations between knowledge density of a country with R&D expenditure, exports and several development indicators, as well as social (e.g., unemployment of educated labor force and NEET, "Not in Education, Employment, or Training") and educational (attainment) indicators. This method, henceforth referred as the Embedding model, is also detailed in Section Models, since we compare it to the Frequentist model.

The predictions for both the Frequentist and the Embedding models are based on the Relative Comparative Advantage (RCA) for each field, which is a normalized metric of productivity. The RCA is widely used to analyze the productivity and specialization level of countries or regions, whether scientific, industrial, technological or commercial [10–16].

Surprisingly, none of these models have been compared in the literature. Each one was tested on a different dataset. In this work, we compare both the Frequentist and the Embedding based models in a systematic way, using a large dataset of publication records from Brazilian researchers collected from Lattes, which is the official platform in Brazil for storing data about researchers [17].

## Studies using Brazilian publication records

The Lattes dataset has been recurrently used by several works in the literature with a variety of objectives in the context of Brazilian science, such as characterizing co-authorship networks

[18], proposing new metrics to quantify the influence of researchers [19], assessing the impact of academic mobility on the quality of graduate programs [20], studying the profiles of top researchers and graduate programs in Computer Science [21, 22], and identifying patterns in interdisciplinary collaborations and analyzing their evolution [23]. The dataset has also been used to demonstrate the effectiveness of the method proposed in [24] for automatic research expertise classification from the titles of academic works. Nevertheless, to the best of our knowledge, this is the first time this data is used to map the Brazilian research space and to understand how this space has changed over time.

## Dynamics of scientific fields

In a recent model proposed by Jaffe *et al.* [6], countries are represented as empirical distributions over research areas, i.e., an array with the proportions for each research area in relation to the total number of publications in such country. The research areas, in turn, can be seen as arrays containing the proportions of the research they represent in each country. For a given set of countries and time frame, the distance between two research areas is measured by the squared Euclidean distance of their array representations. Using a network backbone extraction method, they extract a subgraph containing only the salient relationships between fields. The authors divide countries into strata based on average income data, and then study the research space obtained for each group, their differences and similarities, and how the research space for a given group evolves over time. Their study does not include data from Brazil. On the other hand, since we are only using data from Brazil, their model is not applicable: estimating the distance between areas based on the scientist or institution level granularity would be very noisy since the set of fields in which they are active can be very sparse; on the other hand, estimating it based on the state level granularity would result in only 26 samples. Yet, in our case study, we were able to conduct a temporal analysis similar to that done by Jaffe *et al.* by extracting the backbone of the networks obtained using the Embedding model for Brazil at different time windows.

In an even more recent work, Palmucci *et al.* [25] propose a framework for describing the time evolution of scientific fields, allowing the authors to make predictions about their relative dynamics. The authors use records from the American Physics Society data on articles published during a given period to learn embeddings for PACS codes, which are numerical identifiers used as tags to categorize papers according to physics subfields. The PACS embeddings obtained for subsequent windows allow them to shed some light into the scientific trends in Physics. The authors describe the work as a proof-of-concept and mention that it could be extended to other disciplines. However, it is not clear if this framework could be used to model the relationship between disciplines, as they do not share PACS or other classification codes. Our work differs from theirs in that (i) we are not bound to the Physics domain and (ii) we use the scientists' trajectories to learn about the relationship between fields.

A related approach, based on paper co-authorship, has been used mainly for understanding scientific interactions. The properties and dynamics of co-authorship networks have been well studied [26, 27], and these networks were shown to be especially useful for future performance prediction [28–30]. They have also been used for measuring the strength of the relationship between groups (e.g., countries [31]). In principle, such networks could be used to create research maps by analyzing how researchers from different fields collaborate and even to make predictions regarding the likelihood that an individual becomes active in one field based on past interactions. However, they have not been used for this purpose, to the best of our knowledge.

## Dataset

We use data from the Lattes Platform, which is a vast repository of researchers' curriculum vitae, widely adopted in Brazil. This platform is maintained by the Brazilian National Council of Scientific and Technological Development (CNPq) and is an internationally renowned initiative [17]. The platform is used for several purposes, including assessment of researchers' productivity and quality of academic programs, consequently requiring that all individuals actively pursuing research keep their information up-to-date. We obtain a snapshot of the database collected in early February 2017 using the LattesDataXplorer tool [32]. The vita of each researcher is stored as a XML document.

In what follows, we describe the steps we performed for data acquisition and preparation.

### Step 1: Obtaining venue information

We elect to use the **Scopus field classification scheme**. It contains arguably some of the most important journals for a variety of fields. Each venue is associated with one or more fields. It also has a 3-level hierarchical scheme:

- 4 macro areas (coarsest level),

- 27 intermediate classifications, and

- 308 specific fields (finest level),

which provides useful information for both validation and multi-scale analysis. Fig 1 shows the intermediate classifications grouped by macro areas, identified throughout this manuscript by their acronyms and colors, respectively. We download a list of venues and its associated areas from the SCImago Journal Rank portal [33].

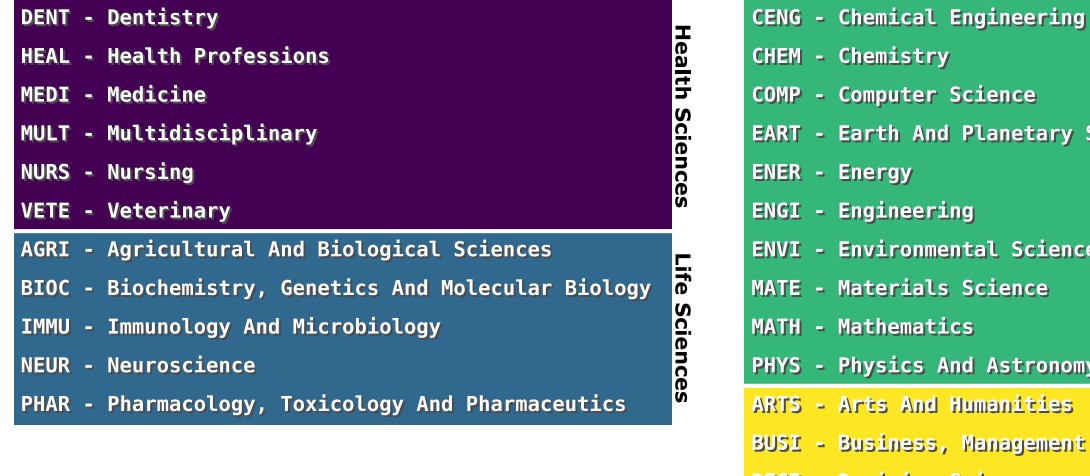

**Fig 1. Scopus' intermediate classification.** List of intermediate classifications for journal labeling according to Scopus, grouped by macro area.

## Step 2: Transforming the data

From each researcher's XML, we extract:

- Personal data: Lattes ID, name of current work place, postal code of current work place.

- Publication data (for every paper): the journal or conference name, the number of authors, year of publication.

The goal of this step is to find the venues in which each researcher publishes. Hence, it is irrelevant whether the same publication appears multiple times, i.e., in different profiles. There is no need to find and treat duplicate occurrences. In addition, by using the researcher's Lattes ID, the data is robust against name changes. We make this dataset available to the research community [34].

## Step 3: Mapping publication data from Lattes to venues and their fields in Scopus

The percentage of papers with perfect matches between venue (journal/conference) names in the Lattes Platform data and the Scopus data is 46%. One of the factors that negatively impacts the number of matches is that, although the researcher can search for venue names when creating or editing his profile, he can also type it manually. Therefore, we adopt an approximate matching described as follows. For each venue in the Lattes Platform, we generate a list containing the substrings obtained when splitting the venue name at the characters in {".", ";", ":", "/", "-"}. For instance, the string *"Example Journal: an experiment/EJ"* yields the list: *"Example Journal: an experiment/EJ", "Example Journal", "an experiment","EJ"*.

When no exact match is found for a venue listed under a Lattes profile, we map it to a journal/conference entry from Scopus by using the first substring from the generated list such that an exact match is found. By doing so, we increased the fraction of covered papers to 52%.

## Step 4: Mapping publication data from Lattes to institutions and states

In addition, we consider all publications of researchers that currently work for one institution as publications of that institution, since we cannot accurately track the past trajectory of researchers. The same principle applies to publications associated with a geographical state. The latter association is done based on the ZIP code of institution as informed in the researcher's profile.

## Limitations

Despite the fact that users can manually enter venue names, we believe that most failures to match are not due to typos, but to venues not present in the Scopus database, such as less known journals and conferences proceedings which are published in Portuguese. This might create a bias in the data, possibly favoring fields with a larger fraction of well-known, international journals.

While the approximate venue matching increases our coverage, it might result in some false positives, and thus add noise to the models. These errors are more likely to occur in the case where a short acronym is present in the name, since venues from different fields can share acronyms. Due to the way the models are built, an incorrect matching would have to occur frequently and systematically in order to perturb them.

One of the issues with the data is inherent to the way it is created. Since the information in each profile is inserted by its owner, there are three main types of errors related to institution names: typos; multiple name variants for the same institution; department names used in *lieu*

of institution names. Instead of attempting to manually fix these errors, we assumed that most researchers enter the official institution name in their profiles.

Last, we associate all the work done by a researcher with the affiliation/state currently listed on his/her profile. We argue that the impact of associating work done by a researcher in a former institution or geographical state with the current one is small, because most of the Brazilian research is developed at public universities and national institutes, where established employees are less likely to change workplace, since it is very hard to do so while keeping their ranks. Reproducing the case study for other countries would require a more precise mapping between papers and affiliations at the time of publication.

## Data summary

Fig 2 shows a simple characterization of the Lattes Platform data. In summary, most of our data is relatively recent: a large number of entries is concentrated in the last 20 years. The fact that the average number of publication records per year grows can either indicate a true growth or a selection bias: the platform was created in 1999, replacing the previous way of collecting Brazilian graduate programs data and thus becoming the standard way of reporting,

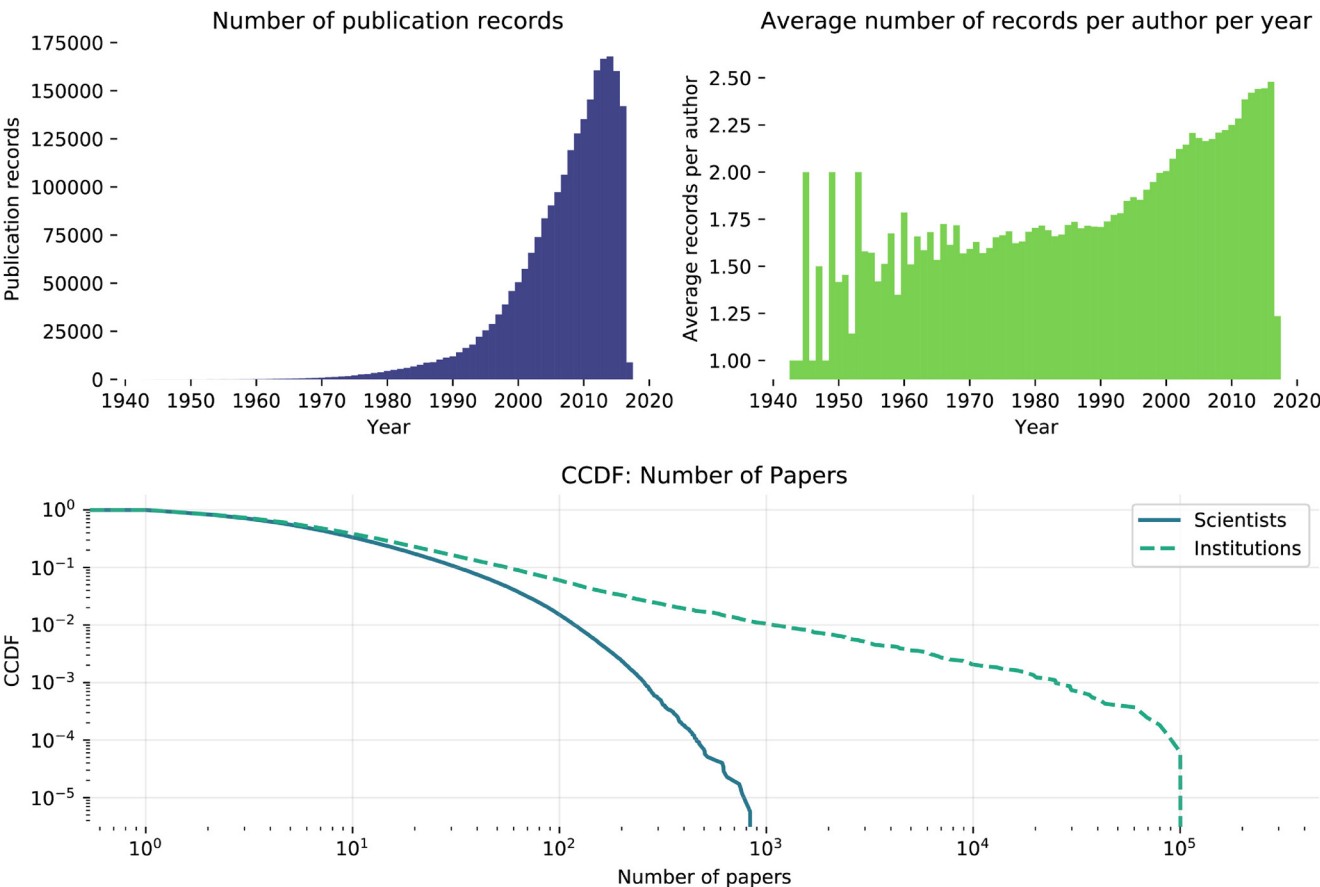

**Fig 2. Data characterization. Top left**: yearly number of new publication entries (a publication is counted multiple times if it is listed under more than one profile). This metric has been increasing since the mid-90s; decrease in last couple of years may be due to right-censoring. **Top right**: yearly number of publications entries normalized by number of authors. Between 1990 and 2016, this number has increased nearly 50%. **Bottom**: Complementary Cumulative Distribution Function (CCDF) of the number of publications per scientist and institution. Both distribution are heavy tailed, but the distribution for institutions seems to follow a power law with cutoff at $x = 10^5$.

**Table 1. Data statistics.** For each macro area *X* shown in the rows, columns report the number of publications entries on venues associated with *X*, number of researchers who authored one or more publications on such venues, the average number of papers published by these authors on those venues and the average number of sub-areas associated with those publication entries.

| | No. Papers Entries *P* | No. Authors *A* | *P/A* | Fields |
|---|---|---|---|---|
| **Health Sciences** | 1078805 (46.6%) | 103886 (59.5%) | 10.38 | 2.41 |
| **Life Sciences** | 1000545 (43.3%) | 97938 (56.1%) | 10.22 | 2.65 |
| **Physical Sciences** | 707327 (30.6%) | 91161 (52.2%) | 7.76 | 2.73 |
| **Social Sci. & Hum**. | 190597 (8.2%) | 56740 (32.5%) | 3.36 | 2.13 |
| **All areas** | 2313151 (100%) | 174717 (100%) | 13.24 | 2.27 |

accessing and evaluating publication records. Last, as observed in research productivity studies using data from other countries [35, 36], the number of publications for scientists in a field is heavy-tailed.

Table 1 shows some statistics of the dataset. We observe that both publications and authors are highly multidisciplinary. The ratio of the first two columns (i.e., the average number of papers a person publishes in a macro area given that she publishes at least one) varies significantly across macro areas, showing that they are quite different. Finally, the last column reflects that the average number of fields associated with a publication does not vary much across macro areas.

## Models

In this section we describe the two state-of-the-art models we use to create research maps, that is, to map the relationships between research areas.

These models consider a scientist (or institution) *s* to be *present* in a research field *f* when his/her normalized number of publications in *f* is greater than a threshold $\theta$. Each publication *p* is normalized by its number of authors $n_p$ and number of fields $m_p$ associated to its venue. Using un-normalized values would give more weight to publications with more authors or to venues that cover multiple fields. An alternative way of discounting these factors would be normalizing by log-scaled values. However, this would result in more transitions and also prevent us from comparing the results with the existing literature.

Hence, we can define the matrix $\mathbf{X}(t) = [X_{sf}(t)]$ over the set of publications $p(s, f, t)$ of scientist *s* in field *f* during time frame *t* as

$$X_{sf}(t) = \sum_{p(s,f,t)} \frac{1}{n_{p(s,f,t)} m_{p(s,f,t)}}.$$

The *presence matrix* $\mathbf{P}(t)$ is then defined by applying a threshold $\theta$ to $\mathbf{X}(t)$:

$$P_{sf}(t) = \begin{cases} 1 & \text{if } X_{sf}(t) > \theta, \\ 0 & \text{otherwise.} \end{cases}$$

Intuitively, the threshold value $\theta$ defines the minimum amount of contribution necessary for an entity to be considered active in a field. We treat $\theta$ as a tuning parameter which must be set by searching the value that maximizes performance in some prediction task.

### The Frequentist model

The first model we use to create research maps is based on a probabilistic approach [5], which we refer as the *Frequentist model*. In this model, the strength of the link between fields *f* and *f'*

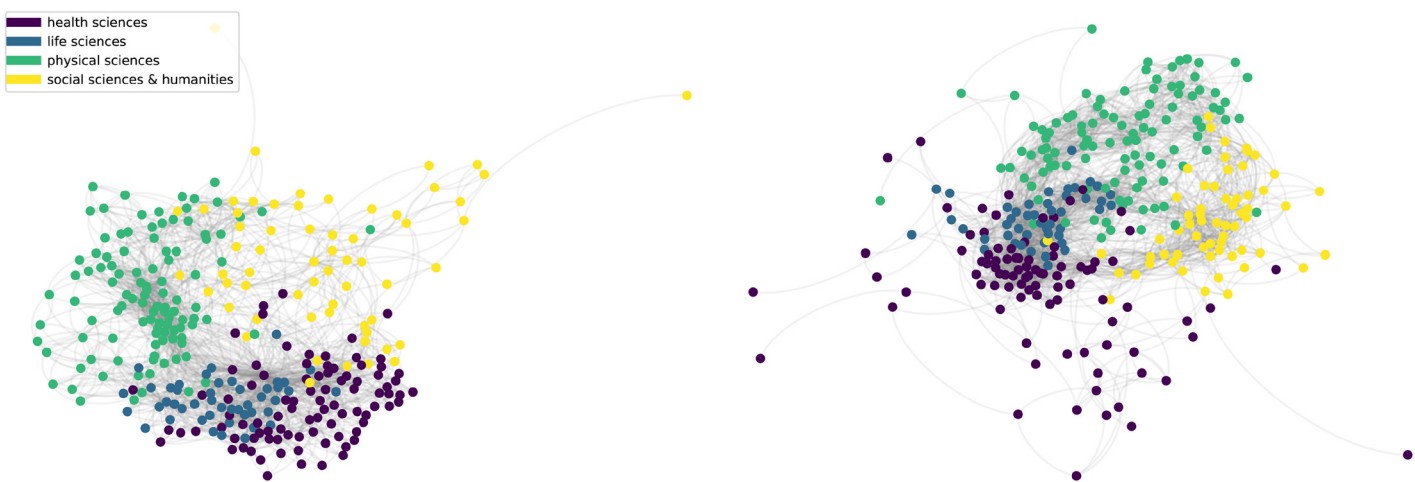

**Fig 3. Graphical representations.** These networks are created using the publication records in [2000, 2014]. Each node is a research field, color-coded according to the respective macro area. We first include the edges of maximum spanning tree and then add all edges whose weights are above a threshold $p$ arbitrarily chosen for better visualization. **Left**: Frequentist model, $p = 0.212$. **Right**: Embedding model, $p = 0.35$.

depends on the probability that a scientist that publishes in $f'$ also publishes in $f$. For this purpose, it is useful to define matrix $\mathbf{M}(t) = [M_{ff'}(t)]$ which counts the number of researchers that published both in $f$ and $f'$:

$$M_{ff'}(t) = \sum_s P_{sf}(t)P_{sf'}(t).$$

Normalizing by the number of scientists that published in $f'$, we obtain the matrix $\phi^{(freq)}(t) = [\phi_{ff'}^{(freq)}(t)]$ of link weights,

$$\phi_{ff'}^{(freq)}(t) = \frac{M_{ff'}(t)}{\sum_s P_{sf'}(t)}.$$

This matrix can be interpreted as the information flow between research fields. Fig 3 (left) depicts the space learned by the Frequentist model for the Brazilian publication data as a network.

### The Embedding model

In another approach, henceforth referred as the *Embedding model*, a *N*-dimensional representation of the relationship data is created [7]. This model is trained on a collaborative filtering based recommendation task, in which each scientist $s$ is represented as the set of fields in which she is present:

$$\mathcal{B}_s(t) = \{f : P_{sf}(t) = 1\}.$$

The model learns vectors $vec_f^t$ for representing each research field $f$ by optimizing an objective function that simultaneously (i) maximizes the similarity between pairs of vectors corresponding to fields present in the same sets $\mathcal{B}_s(t)$ and (ii) minimizes the similarity between those pairs that do not appear together (or rarely appear together). The similarity $S_{ff'}(t)$

between fields $f$ and $f'$ is measured by the cosine similarity:

$$S_{ff'}(t) = \frac{vec_f^t \cdot vec_{f'}^t}{\| vec_f^t \| \| vec_{f'}^t \|}.$$

The same similarity metric is used to map the relationship matrix $\phi^{(emb)}(t) = [\phi_{ff'}^{(emb)}(t)]$ between fields:

$$\phi_{ff'}^{(emb)}(t) = \begin{cases} S_{ff'}(t) & \text{if } S_{ff'}(t) > 0 \\ 0 & \text{otherwise} \end{cases}.$$

As proposed by the authors, we use the algorithm *StarSpace* [8] to obtain the vector representations. Each field $f$ in time frame $t$ is represented by a vector $vec_f^t$ in this space.

Differently than the Frequentist model, the links weights in this model do not have a probabilistic interpretation. Fig 3 (right) shows the space learned by the Embedding model. We observe high modularity with respect to the macro areas.

## Methodology

In this section, we describe the methodology used to quantify the presence of scientist, institution or geographical region in a given field. In addition, we explain how we apply these measures to forecast the next fields in which such entity will become active.

### Revealed comparative advantage

As in previous works [5, 7], we employ the Revealed Comparative Advantage (RCA) [37] to measure the specialization level of a scientist, institution or geographical region in a certain research field. In essence, RCA is the ratio between the fraction of the researcher's publications that fall within a field $f$ and the fraction of all contributions in the data that fall within $f$:

$$RCA_{sf}(t) = \frac{X_{sf}(t)}{\sum_{f'} X_{sf'}(t)} \bigg/ \frac{\sum_{s'} X_{s'f}(t)}{\sum_{s'f'} X_{s'f'}(t)}.$$

We also define discrete stages to study the level of specialization and evolution of a given entity $s$ in a field $f$. When $s$ has no publications in $f$, we label it **inactive** (stage 0); otherwise, we label it **active** (stage $A$), a stage that is further subdivided into **nascent** (stage $N$), when $s$ has few publications in $f$, **intermediate** (stage $I$) when $s$ has some publications in $f$, but still less than the average for $f$, and **developed** (stage $D$), when $s$ has more publications than the average for $f$. Formally,

- Inactive (0): $0 = RCA_{sf}(t)$;

- Active ($A$): $0 < RCA_{sf}(t)$;

  - Nascent ($N$): $0 < RCA_{sf}(t) < 0.5$;

  - Intermediate ($I$): $0.5 \leq RCA_{sf}(t) < 1$;

  - Developed ($D$): $1 \leq RCA_{sf}(t)$.

Fig 4 shows some Brazilian institutions that have specialized in particular fields. For each of the 27 intermediate classifications, we compute the fraction of fields $f$ in which the institution $s$ is developed ($RCA_{sf} \geq 1$).

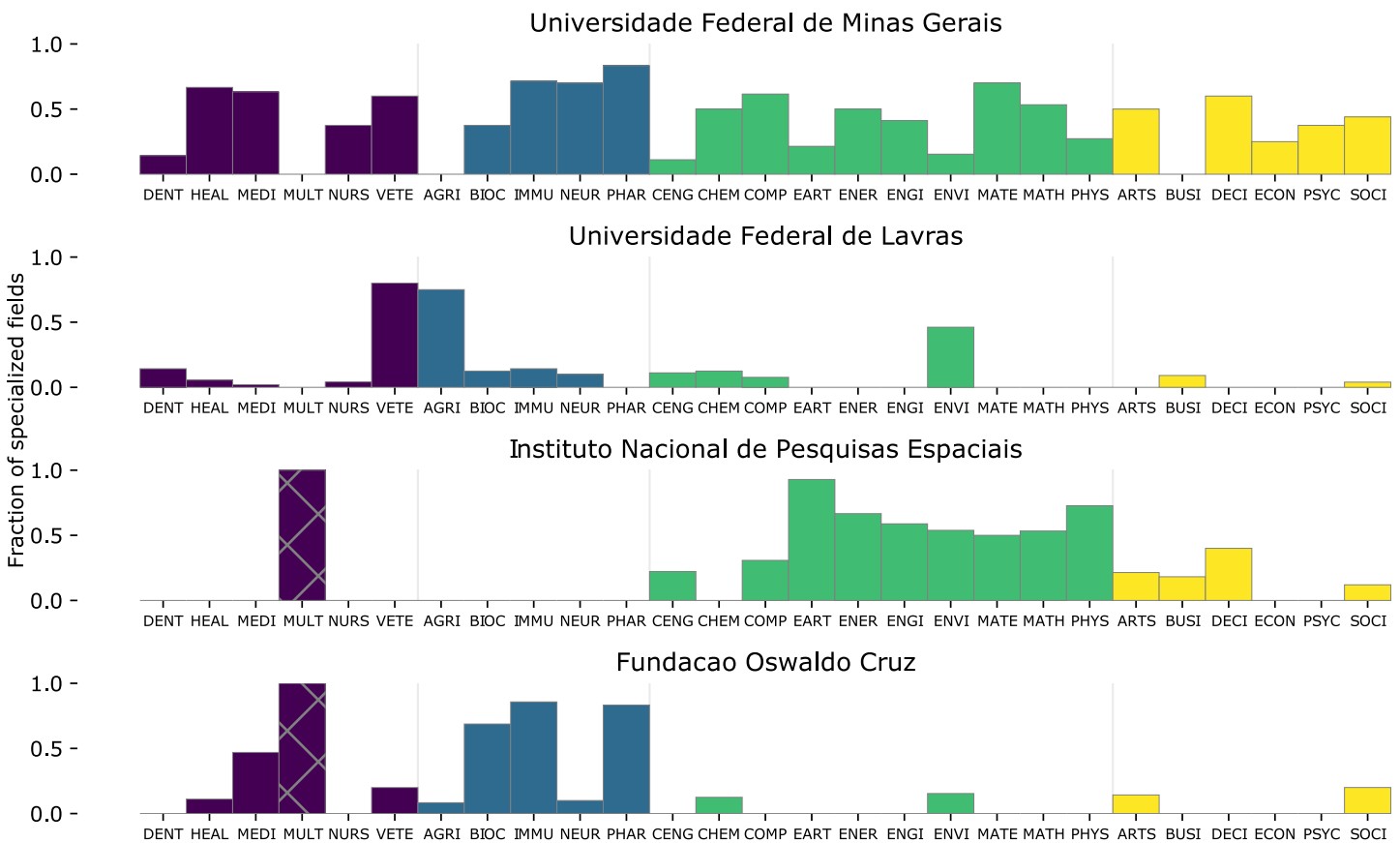

**Fig 4. Institutions specializations.** Fraction of subfields in each of the intermediate fields in which some of the institutions are specialized (data from the time interval [2000, 2014]). Intermediate classifications are grouped and colorcoded according to the macro field. The Multidisciplinary (MULT) intermediate field has only a single subfield and is then shown with hatches.

We observe, for example, that the Universidade Federal de Minas Gerais, the largest federal university in the country, has a very diverse research profile. We also note that the Universidade Federal de Lavras, previously named Escola Superior Agrícola de Lavras (Superior Agricultural School of Lavras), stays highly specialized in the agricultural areas (intermediate classifications VETE, AGRI and ENVI). The Instituto Nacional de Pesquisas Espaciais (National Institute for Space Research) specializes in geosciences, energy, material sciences and other STEM fields. Last, the Fundação Oswaldo Cruz, main reference of the country in the public health, specializes in medicine, microbiology and pharmaceutical fields (intermediate classifications MEDI, BIOC, IMMU and PHAR).

Fig 5 shows that both the distributions of active and developed fields seem to be heavy tailed, although the number of fields imposes a hard cutoff. Not surprisingly, institutions take on a much larger number of areas than individual scientists.

## Transitions prediction

In order to predict the future fields in which a scientist, institution or region *s* will be active, we assume the *Principle of Relatedness* [38–40], which posits that it is easier to specialize in and work on closer fields that require a similar background. There are several works that corroborate this hypothesis [5, 39, 41–47].

**Fig 5. Distribution of number of active and developed fields.** Complementary Cumulative Distribution Function of the number of active/developed fields for scientists and institutions (data from years [2000, 2014]).

We are interested in predicting three types of transition:

- **Scientist becomes Active** $(0 \rightarrow A)$

- **(Active) Scientist becomes Developed**, subdivided into:

  - **Nascent to Developed** $(N \rightarrow D)$

  - **Intermediate to Developed** $(I \rightarrow D)$

Therefore, assuming the Principle of Relatedness, we predict that a scientist $s$ will become active in a field $f$ (transition $0 \rightarrow A$) based on whether or not $s$ is **active** in fields $f'$ related to $f$. Moreover, we predict that $s$ will become developed in a field $f$ (transitions $N \rightarrow D$ or $I \rightarrow D$) based on whether or not $s$ is **developed** in fields $f'$ related to $f$. To predict each of these transitions, we define the appropriate indicator matrix $\mathbf{U}(t) = [U_{sf}(t)]$ over a time window $[t - \Delta_{RCA}, t]$:

$$
U_{sf}(t) = \begin{cases} \mathbf{1}[RCA_{sf}(t) > 0] & \text{for } 0 \rightarrow A, \\ \mathbf{1}[RCA_{sf}(t) > 1] & \text{for } \{N, I\} \rightarrow D. \end{cases}
$$

where $\mathbf{1}[x]$ denotes the indicator function, which equals 1 when predicate $x$ is true, and 0 otherwise; and $\Delta_{RCA}$ is a parameter that determines the time window length. Next, we define the density $\omega_{sf}(t)$, an estimator for the probability that $s$ will further specialize in $f$:

$$
\omega_{sf}(t) = \frac{\sum_{f'} U_{sf'}(t) \phi_{ff'}(t)}{\sum_{f'} \phi_{ff'}(t)},
$$

where $\phi_{ff'}(t)$ is some measure of proximity between fields $f$ and $f'$ in a given space, computed over a time window $[t - \Delta_\phi, t]$. For the space generated by the Frequentist (resp. Embedding) Model, we set $\phi_{ff'}(t) = \phi_{ff'}^{(freq)}(t)$ (resp. $\phi_{ff'}(t) = \phi_{ff'}^{(emb)}(t)$).

We rank all fields $f$ satisfying $U_{sf}(t) = 0$ based on its density $\omega_{sf}$. According to the proximity principle, fields with higher density will transition to a higher development stage before those with lower density. In addition, we compare $\phi_{ff'}^{(freq)}(t)$ and $\phi_{ff'}^{(emb)}(t)$ w.r.t. the quality of their predictions.

### Time windows for model fitting, density estimation and prediction testing

Predicting stage transitions requires the definition of three time windows: (i) one for fitting the model, $[t - \Delta_\phi, t]$; (ii) one for estimating specialization densities, $[t - \Delta_{RCA}, t]$; and one for testing the predictions, $[t + 1, t + \Delta_{test}]$. Note that the model fitting and density estimation windows overlap, but none of them overlap with the prediction testing window. As in previous works, we assume $\Delta_\phi \geq \Delta_{RCA}$ since, intuitively, the relationships between fields change at a much slower rate than a scientist changes fields.

## Experimental results

To evaluate the prediction accuracy of each model, we use the area under the ROC (receiver operating characteristic) curve, obtained by the graphical representation of the relationship between the false positive rate ($x$ axis) and the true positive rate ($y$ axis) for each entity $s$. This area, abbreviated as AUROC, can be interpreted as the probability that a random positive field $f$ is scored higher than a random negative field $f'$ by the model. Hence, an area of 0.5 corresponds to the random baseline. Furthermore, we can only compute the AUROC for entities $s$ that have at least one positive field $f$, i.e., that made at least one transition.

For each model, we compute the AUROC for every scientist, institution and geographical region (Brazil's states) for the relevant transitions. For scientists, we consider only the $0 \rightarrow A$ transition, since there is not enough data for to compute the other transitions. We use different time intervals from the past to generate the research spaces (i.e., $\phi_{ff'}^{(freq)}(t)$ and $\phi_{ff'}^{(emb)}(t)$), and compare the predictions computed for transitions **inactive to active** ($0 \rightarrow A$), **nascent to developed** ($N \rightarrow D$) and **intermediate to developed** ($I \rightarrow D$) with those observed in the data.

Just as in previous works, most of our experiments consist of evaluating the predictions on a recent time window. We follow, as closely as possible, the experimental setups defined in the papers where the Frequentist and Embedding models were proposed. Since they differ, we end up with two main sets of experiments described below. Next, we investigate the impact of institution characteristics and that of the embedding dimension on prediction quality. Last, we evaluate the robustness of the models to long-term predictions.

### First experimental setup

In the first setup, similar to [5], we consider the most recent 3-year window not affected by right-censoring (from 2014 to 2016), the preceding 3-year window for RCA density estimation (from 2011 to 2013) and a longer window for model fitting (from 1999 to 2013). This configuration reoccurs several times in the Experimental Results section. In those cases, we indicate that *time windows are set as in the first experimental setup*.

Recall that the definitions of $\phi_{ff}^{(freq)}(t)$ and $\phi_{ff}^{(emb)}(t)$ depend on the presence matrix $\mathbf{P}(t)$ and, in turn, on the threshold $\theta$. We perform experiments with both models, varying $\theta$ in {0.025, 0.05, 0.10, 0.20, 0.40}. We observe that the prediction performance is robust to the parameter choice (AUROC varies by at most 4%), but that the best results are achieved with $\theta = 0.05$ (in [5], the threshold was fixed to 0.10). Hence, all the results reported in this paper are obtained with $\theta = 0.05$, unless stated otherwise.

Fig 6 shows violin plots of the AUROC for the predicted transitions for scientists, institutions and states, contrasting the accuracy of the two models. Violin plots are similar to box plots in what they show the interquartile interval (vertical bar) and the median (horizontal bar), but they also show the variable's density along the vertical axis. We also include a white circle depicting the average.

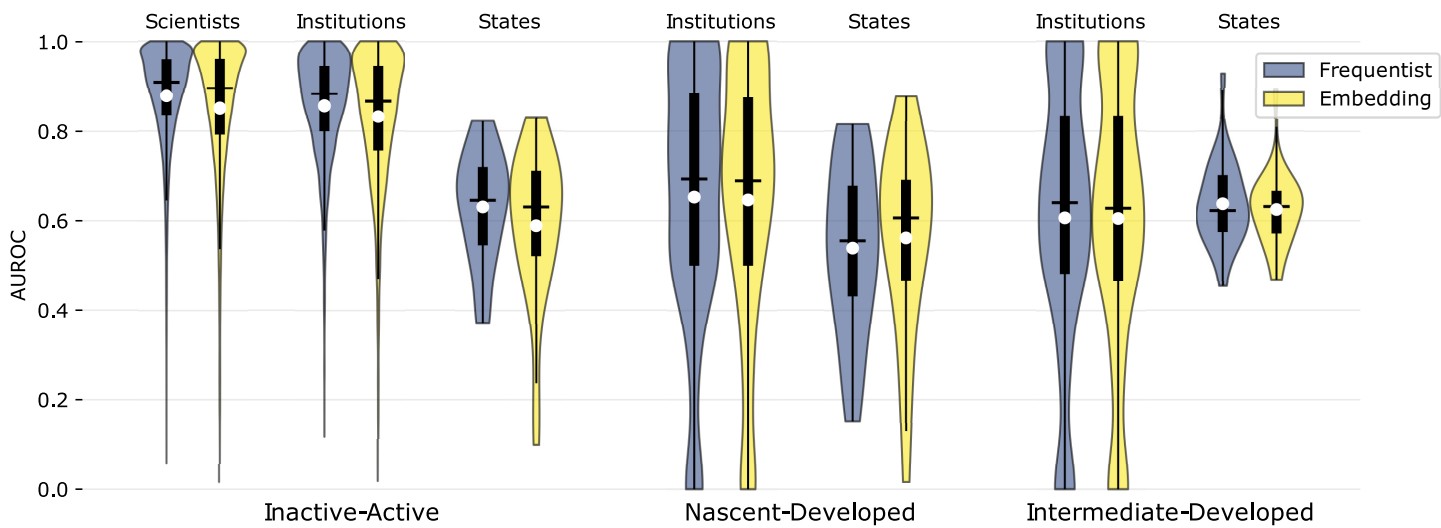

**Fig 6. AUROC's violin plots for all transitions.** Violin plots of the AUROC for the predicted transitions for scientists, institutions and states. For each plot, the black horizontal line shows the median and the white circle shows the mean. Time windows are set as: model fitting, [1999, 2013], density estimation, [2011, 2013], and prediction testing, [2014, 2016].

**Inactive to active ($0 \rightarrow A$).**   We observe that most of these predictions are very accurate for scientists and institutions, but not for states. For scientists and institutions, the empirical distribution of the Frequentist model seem to dominate (in the stochastic sense) that of the Embedding model. For states, the Embedding model even results in some values below 0.3.

**Nascent to developed ($N \rightarrow D$).**   The AUROC for these predictions tend to be much lower than those for the "inactive to active" transition. For institutions, the distribution is concentrated between 0.4 and 1.0 and has median approximately equal to 0.7 for both models. For states, the predictions of the Frequentist model are close to random, while those of the Embedding model have higher median. Overall, the performance of both models when predicting the $N \rightarrow D$ transition is very similar.

**Intermediate to developed ($I \rightarrow D$).**   For institutions, the AUROC for these predictions is slightly lower than that for the "nascent to developed" transition. On the other hand, for states, the predictions are highly concentrated around 0.6.

The corresponding averages are shown in Table 2, where (***) indicates p-value <0.01 in the ANOVA test. We observe that the differences of the averages for transition $0 \rightarrow A$ between the models are statistically significant for both scientists and institutions. Not every scientist, institution and state made a transition during the period between 2014 and 2016, explaining the differences in sample sizes across transitions.

Overall, we observe that both models exhibit similar performance. In fact, the ANOVA test does not discard the hypothesis of equal means for most of the analyzed transitions. The results

**Table 2. Average AUROC for each transition, first setup.** Average AUROC of the predicted transitions for scientists, institutions and states. Time windows are set as: model fitting, [1999, 2013], density estimation, [2011, 2013], and prediction testing, [2014, 2016]. Entities that did not transition during the [2014, 2016] were not included in the sample.

| | $0 \rightarrow A$ | | | $N \rightarrow D$ | | $I \rightarrow D$ | |
|---|---|---|---|---|---|---|---|
| | Scientists | Institutions | States | Institutions | States | Institutions | States |
| **Frequentist** | .879*** | .856*** | .631 | .653 | .539 | .607 | .638 |
| **Embedding** | .851 | .832 | .589 | .646 | .561 | .605 | .625 |
| **Sample size** | 61769 | 5220 | 26 | 441 | 25 | 564 | 26 |

are particularly accurate for the **inactive to active** transition, for both scientists and institutions. The predictions are not as accurate for cases with less data instances, namely those corresponding to other transitions or to the most coarse aggregation (states). A similar decrease in accuracy was observed in [5] for the frequentist model when using the Scopus data.

One difference between our experiment and that proposed in [5] was the inclusion criterion. We include predictions for all valid instances (i.e. all entities that made at least one transition during the testing window), while Guevara *et al.* opted for including only instances whose normalized publication number was above a given threshold, thus focusing only on the most productive scientists and institutions. Our analysis provides a more representative evaluation of the models' performance.

For comparison, we also evaluated the predictions setting the same time windows as the ones used in [5] (model fitting, density estimation and prediction testing set resp. to [1996, 2010], [2008, 2010] and [2011, 2013]). The results obtained are very similar to those with the most recent time window (Table 2).

### Second experimental setup

Differently from the previous methodology, Chinazzi *et al.* [7] evaluate the prediction results for different intervals using 3-year windows for both for RCA estimation and for model fitting. The prediction is then evaluated on the subsequent triennium. Therefore, $\Delta_\phi = \Delta_{RCA} = \Delta_{test} = 2$. The same time intervals were chosen.

This experiment considers transition **inactive to active** $(0 \rightarrow A)$ for scientists. We choose to use boxplots to represent the AUROC distribution in Fig 7, as opposed to violin plots, since the values are highly concentrated between the median and the third quartile, rendering the latter a poor visualization. We observe that AUROC quartiles for a given model remain approximately constant when varying the prediction window from [1989, 1991] to [2007, 2009]. Moreover, the Frequentist model consistently outperforms the Embedding model over time, yielding higher 1st, 2nd and 3rd quartiles for all considered windows. In fact, the lowest

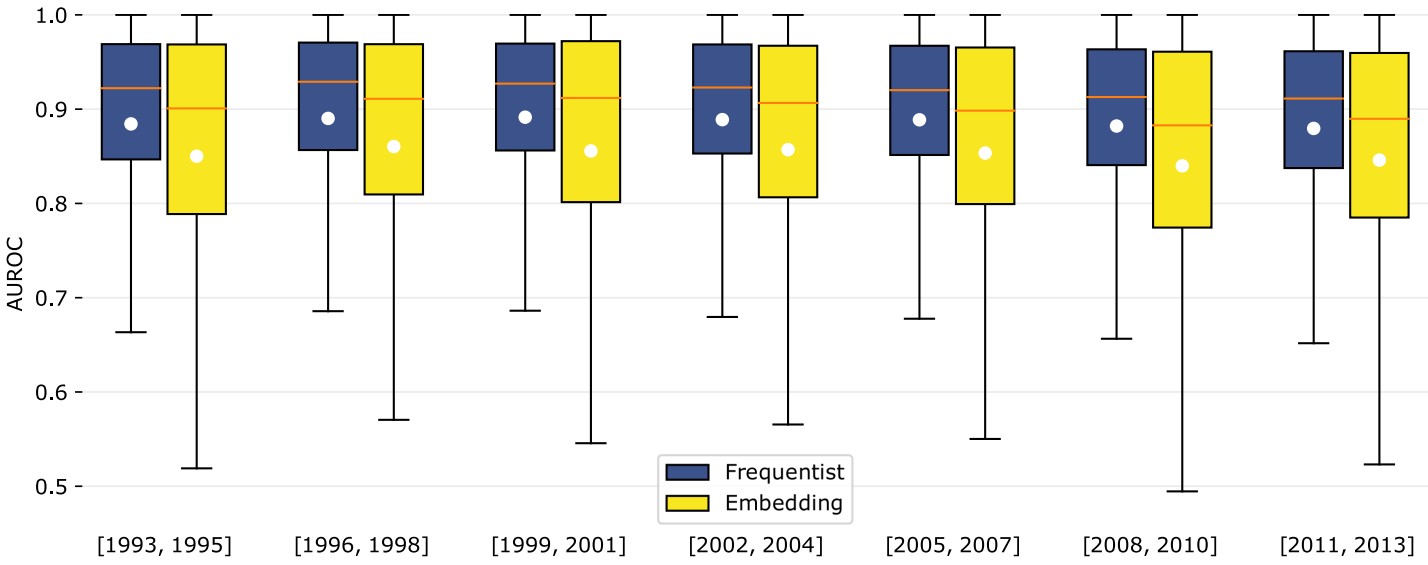

**Fig 7. AUROC's boxplots for scientists' transitions (various periods).** Boxplots of the AUROC for the $0 \rightarrow A$ transition for scientists. The orange horizontal line shows the median and the white circles shows the mean. Time windows for both model fitting and density estimation are shown in the x axis ($\Delta_\phi = \Delta_{RCA} = 2$). Time windows for prediction testing begin immediately after the previous ones ($\Delta_{test} = 2$).

**Table 3. Average AUROC for scientists 0 → A transition, second setup.** Average AUROC of the predicted 0 → A transition for scientists. Time windows for both model fitting and density estimation are shown in the column headers ($\Delta_\phi = \Delta_{RCA} = 2$). Time windows for prediction testing begin immediately after the previous ones ($\Delta_{test} = 2$). Entities that did not transition during the time window were not included in the sample.

| | TIME WINDOWS FOR MODEL FITTING AND DENSITY ESTIMATION | | | | | | |
| --- | --- | --- | --- | --- | --- | --- | --- |
| | **1993-1995** | **1996-1998** | **1999-2001** | **2002-2004** | **2005-2007** | **2008-2010** | **2011-2013** |
| **Frequentist** | .884*** | .890*** | .891*** | .889*** | .889*** | .882*** | .880*** |
| **Embedding** | .850 | .860 | .856 | .857 | .853 | .840 | .846 |
| **Sample size** | 11202 | 16892 | 21879 | 33287 | 44277 | 56241 | 61769 |

median value for the Frequentist model was 0.912 (2007-2009), while the highest median value for the Embedding model was 0.905 (1998-2000).

Table 3 shows the average AUROC of the predicted 0 → A transition for scientists. Once again, (***) indicates p-value <0.01 in the ANOVA test. The difference between the models' performances is statistically significant for all studied intervals. Not all scientists have made transitions in the intervals.

## Impact of statistics of the data associated with an institution on prediction quality

Next, we study how the number of scientific fields and papers associated with an institution affects the prediction quality. We opt for an experimental setup where the number of samples is large, and hence used institutions' data and not scientists' data. As we saw earlier, these properties exhibit a heavy-tailed distribution. In order to understand how they affect our results, we analyze the AUROC as a function of each property. Initially, we expected that fewer fields/papers would lead to lower accuracy, since there is less data to predict transitions.

Fig 8 show contour plots for the empirical distribution of AUROC when predicting transition **inactive to active** (0 → A) for institutions as a function of the (logarithm of the) number of active fields, number of developed fields and (normalized) number of papers (top plots for Frequentist model and bottom plots for Embedding model). We observe that the average AUROC for the "typical institution", i.e., those whose (log-)values of the property under consideration are close to the sample mean. in Brazil is around 0.9. Moreover, the average AUROC decreases with each of the analyzed properties, against our expectations. This is due to the fact that, as the portfolio of active fields in an institution grows, the set of related but inactive fields also grows, turning the prediction task harder.

To understand how each property affects the spread of the AUROC values, we compute the Coefficient of Variation (CV)—defined as the ratio between the sample standard deviation and the sample mean—using a sliding window over the statistic's values. Fig 9 depicts the results using a window of size 1000 (we also tested with window sizes 500 and 2000, but these yielded similar results). The CV decreases with the first two statistics, i.e., when the number of active fields and the number of developed fields in an institution increase Combined with the observations from Fig 8, we conclude that the prediction quality decreases and becomes more concentrated as the number of fields associated with an institution grows. On the other hand, the number of normalized paper entries does not exhibit a clear relationship with AUROC's CV. As noted in the previous sections, the quality of the predictions made by Frequentist model is more concentrated around the average than those of the Embedding model.

## Robustness of models for long-term predictions

We also evaluate the robustness of the models when predicting transitions from **inactive to active** (0 → A) made by scientists further in the future. In this experiment, we shift back the

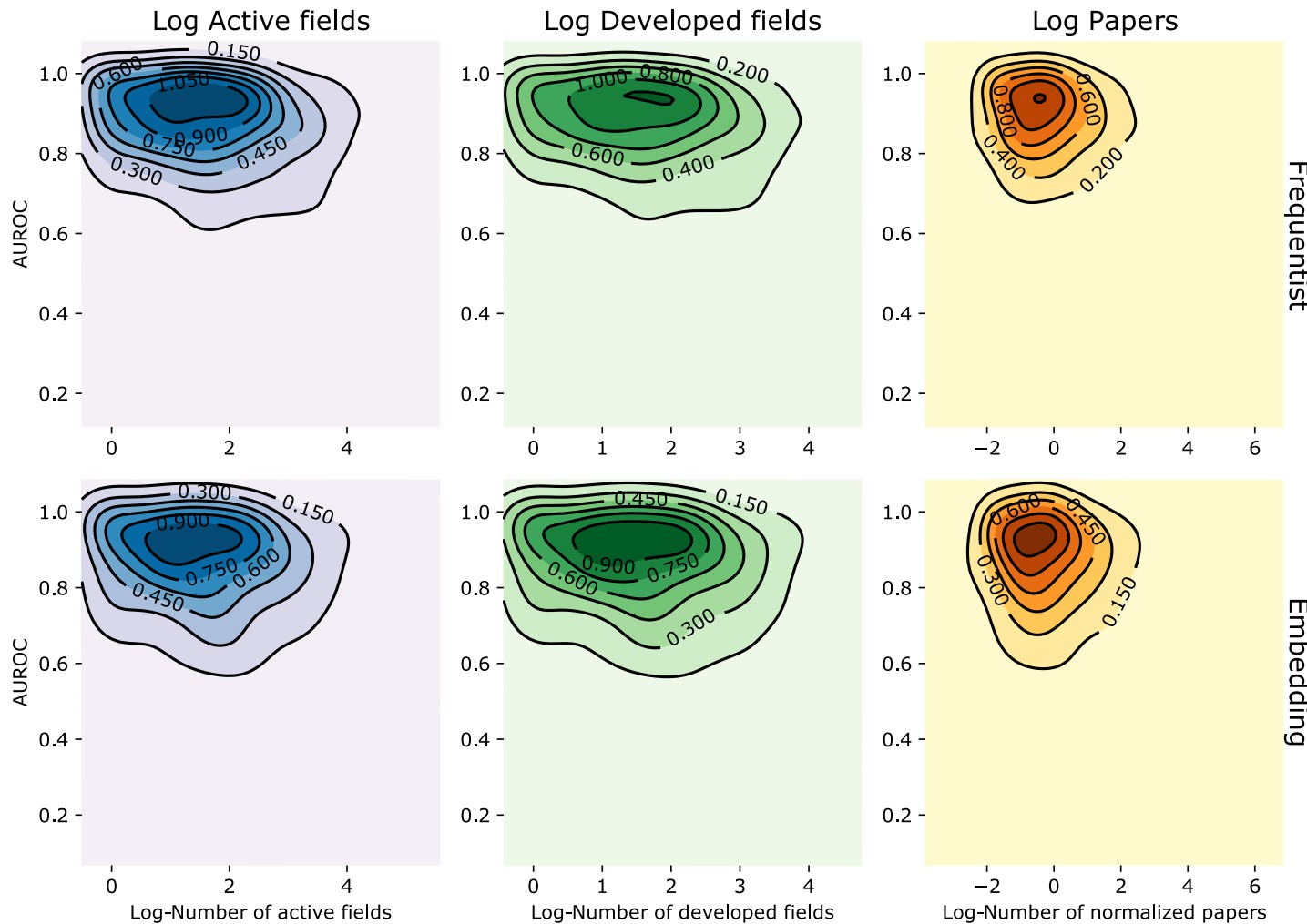

**Fig 8. AUROC distribution for institutions vs. data properties.** Empirical distributions of the AUROC as a function of the logarithms of the (i) number of active fields, (ii) number of developed fields and (iii) normalized number of papers. Time windows are set as in the first experimental setup. Contour plots were created from 500 sample points using Gaussian kernels.

model fitting and density estimation windows resp. to [1990, 2004] and [2002, 2004] so that we can evaluate several testing windows. More precisely, we use all 3-year windows starting between 2005 and 2014 for testing the predictions. Fig 10 shows the 1st, 2nd and 3rd quartiles of the AUROC values for each testing window for the Frequentist model (top) and Embedding model (bottom) as separate plots to better visualize the AUROC variation for each model. Both models are shown to be robust w.r.t. shifts in the prediction window, although the Frequentist model displays less performance degradation with time. One possible explanation for the robustness of the models is that activity in new fields in the near future can persist during the following years.

In addition, we note that the AUROC is very high for scientists during the considered period. The first quartile for the [2005, 2007] period is close to the median for the [2014, 2016] period in the first experimental setup. We emphasize that only the scientists that made at least one transition for each of the test windows (considering the set of fields in which they were active during the [2002, 2004] interval) were taken into account, which may incur a bias towards more productive scientists.

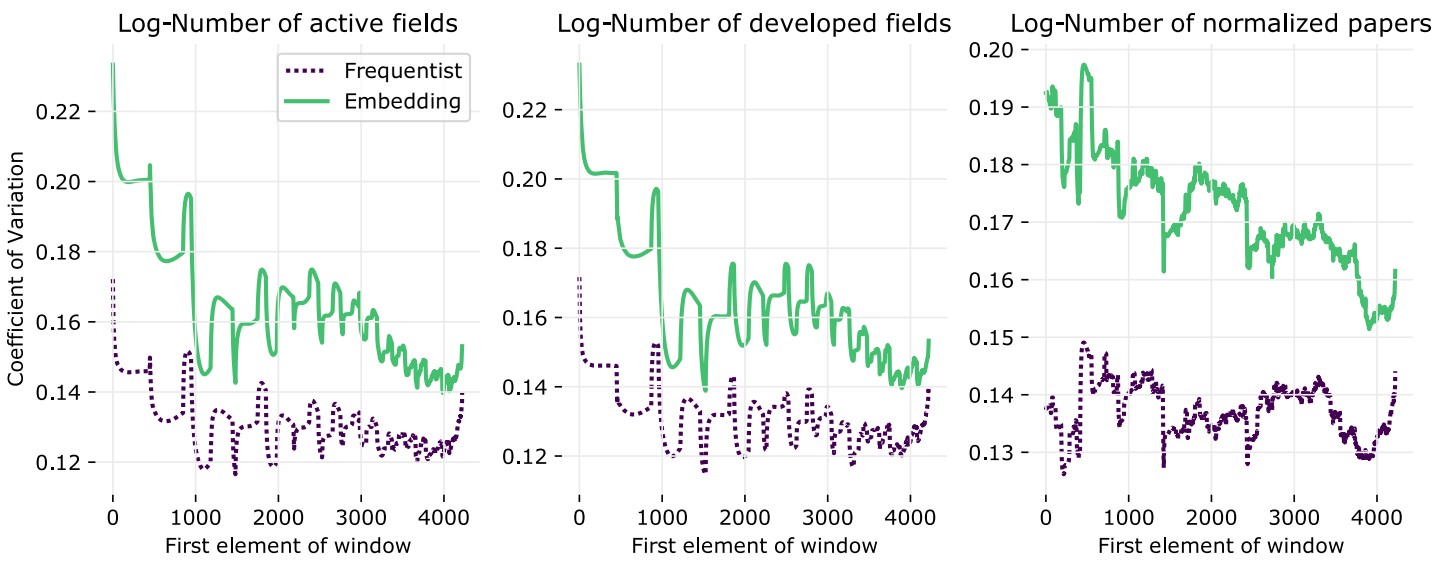

**Fig 9. AUROC's Coeff. of variation for institutions vs. data properties.** Ratio between sample standard deviation and sample mean of the AUROC as a function of the (i) number of active fields, (ii) number of developed fields and (iii) normalized number of papers. Time windows are set as in the first experimental setup. Statistics are computed using a sliding window of size 1000 on the *x* axis.

## Case study: Mapping the Brazilian research space

In this section, we conduct an in-depth analysis of the research spaces created by each model from the Lattes dataset of publication records. This analysis is two-fold: first, we analyze

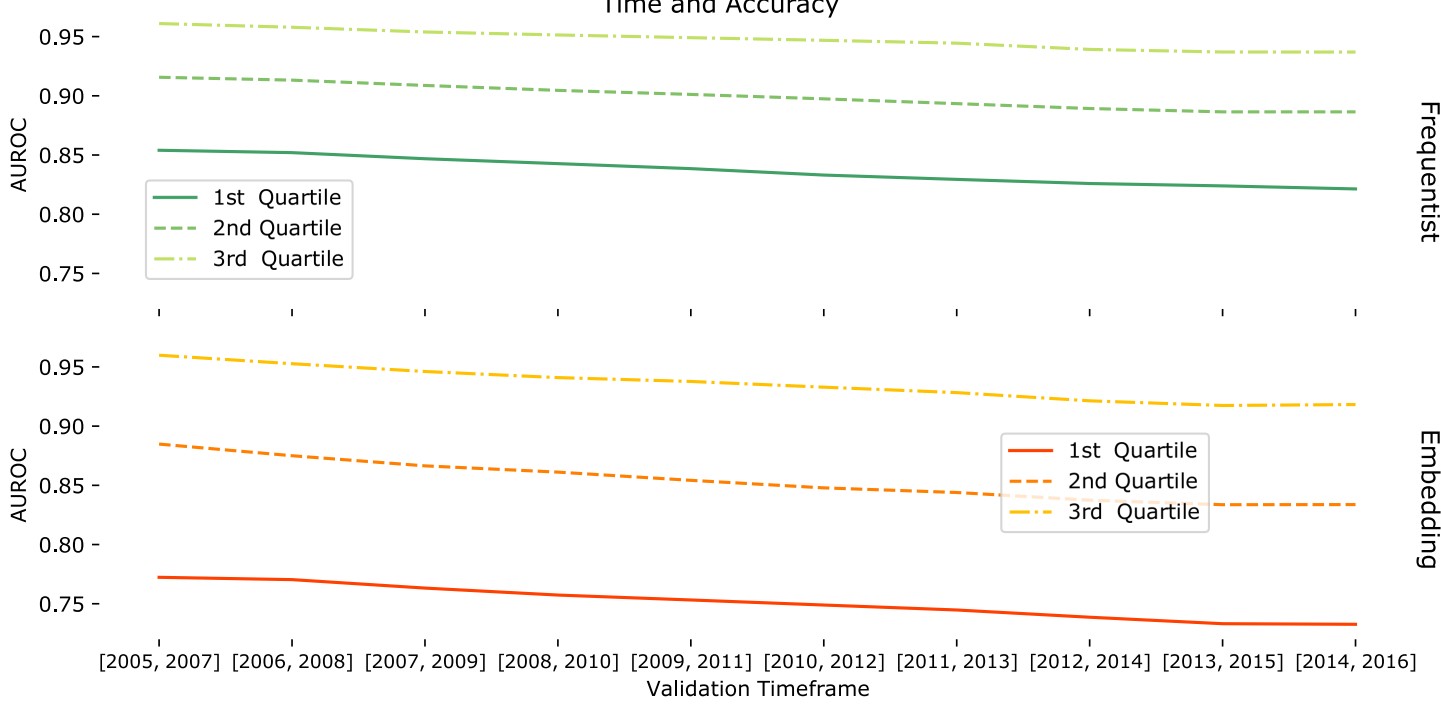

**Fig 10. Long-term predictions.** First, second and third quartiles of AUROC yielded by the Frequentist model for different prediction testing windows, when predicting the transition **inactive to active** for the 15,491 scientists that performed some transition in every window. Time windows for model fitting and density estimation are resp. [1990, 2004], [2002, 2004]. We consider all 3-year windows starting between 2005 and 2014 for testing the predictions.

predictions for well-known Brazilian scientists, relying on some knowledge about the careers of these individuals to understand when the model makes good predictions and when it fails; second, we analyze the research spaces using backbone and community extraction algorithms to understand how the panorama of research fields in Brazil has changed over the years.

## Analyzing individual predictions for scientists

Looking into the predictions made for individual scientists can give us a better understanding of the scenarios where the models work and those where they fail, including what kinds of mistakes they make. Here we choose two researchers somewhat arbitrarily in favor of the convenience of conferring with them (or with someone that knows them well enough) to reason why models are making certain predictions.

We compute the predictions made by the Frequentist and Embedding models for professors Jussara Marques de Almeida and Renato Martins Assunção, two top tier computer science researchers in Brazil. The tables below show, for the respective scientists, the models' rankings, containing the top 10 predicted new fields, and the "actual ranking", containing the top 10 fields joined by the researcher in [2014, 2016] sorted by RCA (higher to lower). Overall, both models predicted research fields that were close to the true ones, but they rarely return the exact right fields.

For Jussara M. Almeida (Table 4), the Frequentist model is able to make three exact predictions of top 10 ("hardware and architecture", "human-computer interactions" and "media technology"), while the Embedding model could predict only one ("hardware and architecture"). Once again, we observe some predictions that are closely related to the actual fields joined by her: the frequentist model predicts "management information systems", which is obviously close to "information systems and management" and to "information systems". Here, the Frequentist model is arguably the best of the two.

The trajectory of Renato M. Assunção (Table 5), defines a particularly challenging prediction task. In 2013, Prof. Assunção transferred from Department of Statistics to the Department of Computer Science at the Universidade Federal de Minas Gerais. He is a statistician whose early work is mainly focused on geospatial models, but with some contributions in accounting and econometrics. Since 2013, he started publishing in computer science venues, particularly those related to machine learning and data mining, which was not predicted by the models. Both models predict that he would join fields that are close to those of his early work: "analysis", "geometry and topology". And also both predicted work in computer science and in

**Table 4. Jussara Marques de Almeida.** Top 10 predicted areas for inactive to active transition for professor Jussara Marques de Almeida, using both models, and actual top 10 by RCA. Time windows are set as in the first experimental setup.

| Frequentist | Embedding | Actual |
|---|---|---|
| logic | **hardware and architecture** | **human-computer interaction** |
| **hardware and architecture** | signal processing | computer networks and communications |
| **management information systems** | control and optimization | **media technology** |
| signal processing | computational theory and mathematics | **information systems and management** |
| computer vision and pattern recognition | electrical and electronic engineering | theoretical computer science |
| **human-computer interaction** | statistics, probability and uncertainty | artificial intelligence |
| **media technology** | modeling and simulation | information systems |
| e-learning | **library and information sciences** | **hardware and architecture** |
| control and optimization | geometry and topology | **library and information sciences** |
| computer graphics and computer-aided design | **management information systems** | computer science applications |

**Table 5. Renato Martins Assunção.** Top 10 predicted areas for inactive to active transition for professor Renato Martins Assunção, using both models, and actual top 10 by RCA. Time windows are set as in the first experimental setup.

| Frequentist | Embedding | Actual |
| --- | --- | --- |
| algebra and number theory | geometry and topology | epidemiology |
| geometry and topology | algebra and number theory | statistics and probability |
| analysis | analysis | cognitive neuroscience |
| nature and landscape conservation | parasitology | **computer science (miscellaneous)** |
| **computational mathematics** | cultural studies | artificial intelligence |
| mathematics (miscellaneous) | history and philosophy of science | instrumentation |
| control and optimization | **mathematics (miscellaneous)** | **computer science applications** |
| **mathematical physics** | microbiology (medical) | **atomic and molecular physics, and optics** |
| statistical and nonlinear physics | **theoretical computer science** | analytical chemistry |
| computational mechanics | nature and landscape conservation | biochemistry |

physics, but in different areas. In sum, both models had weak performances and could not predict such a sharp change of research areas.

In general, we observe that the models fail to make good predictions for individuals who become active in an area in which interest has spiked too abruptly. Moreover, as expected, none of the models can accurately predict transitions made by someone who changed fields. Although the fields in the actual rankings were rarely predicted by the models, we observed that many of the predicted fields are closely related to the actual ones. In conclusion, this analysis characterizes some types of errors made by the models for individuals' predictions. It also suggests that evaluation metrics which account for the relatedness between areas should be considered in future works, as they may be better capable of discriminating the quality of the research maps.

## Network backbone extraction

So far we have considered the finest level of the Scopus 3-level hierarchy, consisting of a large number of highly connected research fields. Although this is a good choice for learning research spaces that can generate fine level predictions, it makes it very difficult to observe trends in the dynamics directly from the networks. To better analyze this dynamics, we group fields using the intermediate level of the Scopus hierarchy, rendering 27 classifications. In addition, we prune less relevant edges using a network backbone extraction algorithm as described below.

Here we reproduce the methodology proposed by Jaffe *et al.* [6] for analyzing the evolution of research fields' relationships using our data. Since the space generated by the Embedding model is symmetric and the weights between research fields are non-negative, we can apply the *Disparity Filter* [48] algorithm to the resulting research space in order to select only the edges whose weight is statistically significant, given a significance threshold $\alpha$. As the Frequentist model generates a directed graph, we cannot replicate this methodology with that model. Differently than in [6], our graph depends on a threshold $\theta$ used for defining the presence matrix. Note that there is an interplay between $\theta$ and $\alpha$, but there is no way to automatically tune them when conducting a qualitative analysis of the resulting backbone network. Hence, we perform experiments with different values of thresholds, ultimately setting $\theta = 0.10$ and $\alpha = 0.20$ (which is higher than the $\alpha = 0.05$ value used in [6]) since, in our case, these values yield networks that are neither too dense or too disconnected to be interpreted.

Continuing to follow the aforementioned methodology, we apply a greedy community detection algorithm [49] on the backbone network to find out how the different research fields are grouped. Fig 11 shows the results when fitting the Embedding model to the following

**Years 2003-2007**

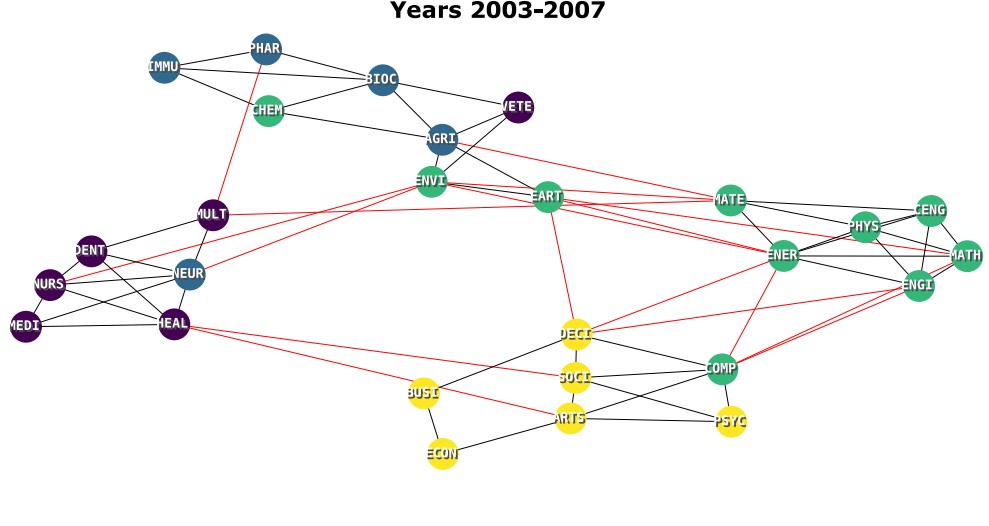

**Years 2008-2012**

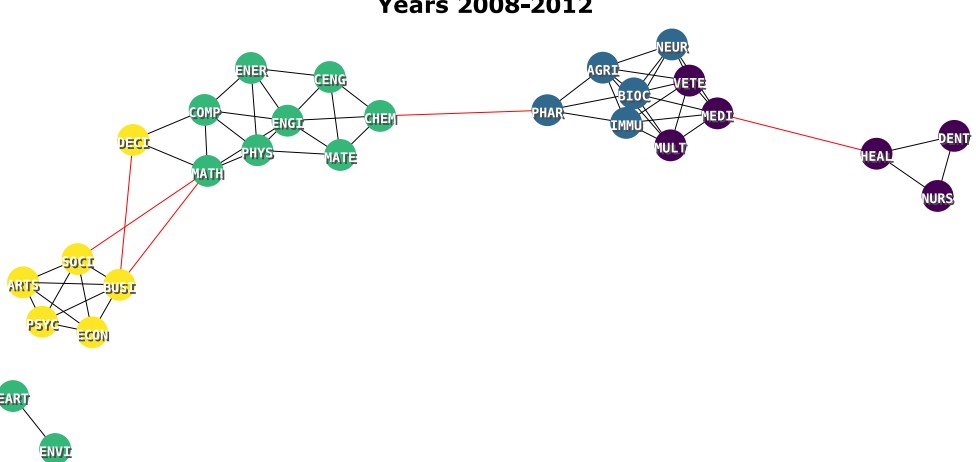

**Years 2012-2016**

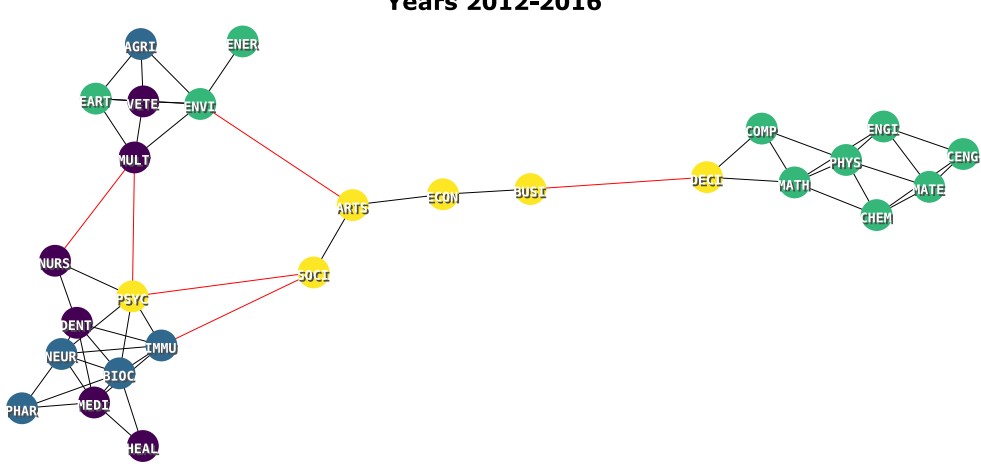

**Fig 11. Backbone network.** Backbone with $\alpha$ = 0.20 of the space created by the embedding model for the intermediate areas, for different time intervals. The color of the edge represents whether it is inter-group (**red**) or intra-group (**black**). The color of the vertex represents the associated macro area.

5-year data windows: 2003 to 2007, 2008 to 2012, and 2012 to 2016. The first two windows are the same as in [6]), but we needed to shift the third window to the left by a year, as our data was collected in 2017. This allows for a better comparison with [6] as opposed to shifting all windows. Intra-group and inter-group edges are respectively shown in black and red. Vertex colors indicate one of the four classifications in the highest hierarchy level: Health Sciences (purple), Life Sciences (blue), Physical Sciences (green) and Social Sciences and Humanities (yellow). Some congruence between these classifications and the obtained groups is expected. A visual inspection reveals that:

- Physics (PHYS), Mathematics (MATH) and other STEM areas (yellow) are consistently close, even though they are occasionally allocated to different groups;

- A similar observation can be made about the Social Sciences and Humanities (yellow);

- Life Sciences (blue) and Health Sciences (purple) macro areas have a lot of interaction and their separation is not always clear;

- Earth sciences (EART) and Environmental Sciences (ENVI), which together make up a component in the [2008, 2012] range, are linked in all periods;

- Agriculture and Biological Sciences (AGRI) and Veterinary (VETE) are consistently linked. They form a clique with Biochemistry (BIOC) (i.e., a triangle, in the case of three vertices) in the first two windows;

- Immunology (IMMU) and Pharmacology (PHAR) are consistently close, but no longer linked in the last window;

- The Health Professions (HEAL), Dentistry (DENT) and Nursing (NURS) clique occurs in the first two intervals, but break up in the last;

- In the first two intervals, Chemistry (CHEM), Pharmacology (PHAR) and Biochemistry (BIOC) form a consistent path, but break up in the third interval;

- From the first to the second interval, the number of edges between groups decreases and that of intra-groups increases;

- There is a reduction in the total number of statistically significant edges through the time windows (65, 60 and 54 edges, respectively).

We contrast these observations with the those from the existing body of work, revealing many similarities despite the use of different methodologies. For instance, Pessoa Junior *et al.* [23] studies Brazilian interdisciplinary collaborations based on the self-declared classifications of researchers into 8 major areas defined in Lattes. Although their results are not directly comparable to ours because (i) we use the Scopus classification scheme and (ii) they consider publication records starting from 1952, we can pinpoint some common findings, such as the link between Social Sciences and Humanities and the link between Life Sciences and Biological Sciences.

Jaffe *et al.* [6] propose a model that maps the research space based on the proportion that each of the research areas represent in the total research output of different countries. Although the model differs from the ones used here, we can still explore similarities and differences in the networks they yield. More specifically, we consider the backbone networks they created by grouping countries based on income bracket for different times periods. We compare those networks with the ones we obtain for Brazil to understand which aspects of

Brazilian research resemble those of developed countries and which ones are closer to those of developing countries.

**Similarities.**

1. There is consistency in the grouping of STEM areas (yellow);

2. Upper-middle income countries consistently exhibit a link between Agriculture and Biological Sciences (AGRI) and Biochemistry (BIOC), as in our two first windows;

3. Upper-middle and high income countries exhibit a link between Earth sciences (EART) and Environmental Sciences (ENVI) in the final two time frames;

4. For upper-middle and high income countries, the life sciences (blue) and health sciences (purple) areas display a lot of interaction.

**Differences.**

1. The networks in Brazil are less dense;

2. They have a larger number of components;

3. For upper-middle and high income countries, the social sciences and humanities fields (yellow) are grouped together, but not as consistently as in our data (sometimes they split). Furthermore, for these countries, Decision Sciences (DECI) is always in other groupings.

We observe that Brazil presents similarities with both upper-middle and high income countries, but also exhibits a link between Agriculture and Biochemistry that is exclusive of upper-middle countries. This is not unexpected, given that Brazil is indeed an upper-middle income country. Yet, this indicates a high degree of consistency between the two models (i.e., the Embedding model and the one in [6]), in spite of fundamental differences in the way they are constructed and in the data used to obtain the research spaces.

The first two differences are sensitive to the choice of the threshold $\alpha$: choosing a smaller $\alpha$ would make the networks denser and with a lesser number of connected components. Therefore it cannot be concluded that research fields in Brazil are less connected (or more isolated) from the first two differences. Regarding the third difference, these areas are known to exhibit strong collaboration ties in Brazil due to the similarity in the way they work [23].

## Conclusions

To the best of knowledge, we performed the first systematic evaluation of the state-of-the-art methods for creating research maps, referred in the text as the Frequentist model [5] and the Embedding model [7]. For this purpose, we used data collected from the Lattes Platform, which is the official platform in Brazil for storing data about researchers. The two main sets of experiments presented here aim to reproduce the methodologies defined in [5] and [7] as closely as possible.

The first experimental setup considers the task of predicting transitions for a recent 3-year window (2014-2016). In this task, both models achieved good results when predicting the **inactive to active** ($0 \rightarrow A$) transition for scientists and institutions (average AUROC $>0.8$), but the Frequentist model outperformed the Embedding model by approximately 3-4%. The differences in AUROC were statistically significant (p-value $<0.01$). On the other hand, the two models performed poorly and were statistically tied when predicting the $0 \rightarrow A$ transition for states (average AUROC $\approx 0.6$), and the **transitions to developed stage** ($N \rightarrow D$ and $I \rightarrow D$) for institutions and states (average AUROC $\in [0.54, 0.65]$). We conjectured that the lower

performance observed for the latter transition types is due to the number of samples (especially low in the case of states) and to the fact that more aggregation implies in a larger number of active fields and, hence, in a larger number of potential fields to be developed.

The second experimental setup considers transition **inactive to active** ($0 \rightarrow A$) for scientists over various testing windows. Once again both models yielded good results. Moreover, we observed that the AUROC quartiles for a given model remain approximately constant when varying the prediction window from [1996, 1998] to [2014, 2016]. In particular, the Frequentist model consistently outperforms the Embedding model over time, yielding higher 1$^{st}$, 2$^{nd}$ and 3$^{rd}$ quartiles for all considered windows.

We investigated the impact of statistics of the data associated with an institution characteristics on the prediction quality. We observed that the average performance decreases with the number of active fields, developed fields, and number of papers an institution has. This corroborates our conjecture that having more active areas turns the prediction task more difficult.

We have also shown that both models are robust enough to make predictions further in the future: using [1990, 2004] as the fitting interval and [2002, 2004] as the RCA estimation interval, the median AUROC decreased only by 4% when shifting the prediction window 9 years into the future (from 2005-2007 to 2014-2016).

In addition, we conducted an in-depth analysis of the research spaces created by each model from the Lattes data. We used predictions made for two prolific scientists to understand when and how the models fail. Not surprisingly, models fail to make good predictions for individuals who become active in an area in which interest has suddenly spiked, as well as for individuals who have changed fields. On the positive side, some of the incorrect predictions are close to the actual fields to which a researcher has transitioned, suggesting that evaluation metrics that consider the relatedness between areas can be proposed to better discriminate the performance of research space models.

Last, we analyze the dynamics of the Brazilian science over three consecutive time windows—2003 to 2007, 2008 to 2013 and 2012 to 2016—by extracting the backbone networks of the research spaces generated for each period. We compare the results with the networks in [6]. Among the key observations, we notice that Brazil presents similarities with upper-middle income countries networks, which were uncovered using a completely different model and data, thus corroborating our results.

## Acknowledgments

The authors sincerely thank Thiago M. R. Dias, who has kindly provided the Lattes dataset used in this work, and Alberto H. F. Laender, for his helpful comments on the present research.

## Author Contributions

**Conceptualization:** Francisco Galuppo Azevedo, Fabricio Murai.

**Data curation:** Francisco Galuppo Azevedo.

**Formal analysis:** Francisco Galuppo Azevedo, Fabricio Murai.

**Investigation:** Francisco Galuppo Azevedo.

**Methodology:** Fabricio Murai.

**Resources:** Fabricio Murai.

**Software:** Francisco Galuppo Azevedo.

**Supervision:** Fabricio Murai.

**Visualization:** Francisco Galuppo Azevedo.

**Writing – original draft:** Francisco Galuppo Azevedo, Fabricio Murai.

**Writing – review & editing:** Francisco Galuppo Azevedo, Fabricio Murai.

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
