## [Decision Letter · Decision Letter 0]

5 Jan 2021

PONE-D-20-37287

Evaluating the state-of-the-art in mapping research spaces: a Brazilian case study

PLOS ONE

Dear Dr. Murai,

Thank you for submitting your manuscript to PLOS ONE. After careful consideration, we feel that it has merit but does not fully meet PLOS ONE’s publication criteria as it currently stands. Therefore, we invite you to submit a revised version of the manuscript that addresses the points raised during the review process.

We look forward to receiving your revised manuscript.

Kind regards,

Chi-Hua Chen, Ph.D.

Academic Editor

PLOS ONE

Journal Requirements:

Reviewers' comments:

Reviewer's Responses to Questions

**Comments to the Author**

1. Is the manuscript technically sound, and do the data support the conclusions?

Reviewer #1: Yes

Reviewer #2: Yes

Reviewer #3: Partly

2. Has the statistical analysis been performed appropriately and rigorously? 

Reviewer #1: I Don't Know

Reviewer #2: Yes

Reviewer #3: N/A

3. Have the authors made all data underlying the findings in their manuscript fully available?

Reviewer #1: Yes

Reviewer #2: Yes

Reviewer #3: Yes

4. Is the manuscript presented in an intelligible fashion and written in standard English?

Reviewer #1: Yes

Reviewer #2: Yes

Reviewer #3: Yes

5. Review Comments to the Author

Reviewer #1: The authors describes an interesting field of connecting Research Areas am mapping them.

The paper is well written and provides interesting Information how science works in a defined area.

However, the paper is very long and should be focussed on its main statement.

The abstract should be revised and the citations (Guervarra et al., Chinazzi et al, etc.) should be deleted or transfered to Introduction. Readers who aren´t expert of this theme would be distracted from the main focus of this work.

Please describe the abbreviations sufficiently (DoD, UCSD, word2vec, NEET).

Could the authors provide more information regarding scientist with a reasaerch stay abroad. Are These works integrated in this model?

What happens if a scientist changes her/his name after marriage? Are these changes considered in the methology?

Reviewer #2: In this manuscript, a comparative analysis of two methods to evaluate the ability of them to predict the potential of some researchers to do contribution in others fields.

The idea presented is interesting, but I don't know the necessity to perform this comparison. Moreover, authors don't take into account that the fields are not disjoint. That is, some publications are classified into two or more fields. For example, Scientometrics could be seen as computer science, and library and information science, among other disciplines. For example, the journal of Scientometrics, is indexed in two categories. So, researchers publishing in this journal, have presence in both areas. This issue is important, and could affect the main hypothesis of the paper.

In what follows, some comments and suggestions are listed:

- I miss references and comments regarding science mapping analysis and bibliographical networks. Authors should cite too small when they talk about co-citation, and do the same with coupling and other networks.

- There are more ways to create a map, for example, based on author information, that is, a co-authors networks (author name, or affiliation).

- The dataset employed is old, and not accurate. Authors only match 52% of the publications. Maybe it could e better use directly Scopus as databased, and join by Scopus author ID.

- The figures have a very low quality. Very low.

Reviewer #3: This article is very well written. However, I do have multiple comments that I believe can help improve it. These are described in detail in the attached document.

6. PLOS authors have the option to publish the peer review history of their article (what does this mean?). If published, this will include your full peer review and any attached files.

Reviewer #1: No

Reviewer #2: No

Reviewer #3: No

---

## [Author Response · Author response to Decision Letter 0]

22 Feb 2021

The response to the reviewers has been uploaded as an attachment, as it is in slight excess of 20,000 characters.

---

## [Decision Letter · Decision Letter 1]

4 Mar 2021

Evaluating the state-of-the-art in mapping research spaces: a Brazilian case study

PONE-D-20-37287R1

Dear Dr. Murai,

We’re pleased to inform you that your manuscript has been judged scientifically suitable for publication and will be formally accepted for publication once it meets all outstanding technical requirements.

Kind regards,

Chi-Hua Chen, Ph.D.

Academic Editor

PLOS ONE

Additional Editor Comments (optional):

Reviewers' comments:

Reviewer's Responses to Questions

**Comments to the Author**

1. If the authors have adequately addressed your comments raised in a previous round of review and you feel that this manuscript is now acceptable for publication, you may indicate that here to bypass the “Comments to the Author” section, enter your conflict of interest statement in the “Confidential to Editor” section, and submit your "Accept" recommendation.

Reviewer #1: All comments have been addressed

Reviewer #2: All comments have been addressed

2. Is the manuscript technically sound, and do the data support the conclusions?

Reviewer #1: Yes

Reviewer #2: Yes

3. Has the statistical analysis been performed appropriately and rigorously? 

Reviewer #1: I Don't Know

Reviewer #2: Yes

4. Have the authors made all data underlying the findings in their manuscript fully available?

Reviewer #1: Yes

Reviewer #2: Yes

5. Is the manuscript presented in an intelligible fashion and written in standard English?

Reviewer #1: Yes

Reviewer #2: Yes

6. Review Comments to the Author

Reviewer #1: The authors were able to answer all of my comments. I think this study may add new aspect in the field of scientific networks. However, with regard to the calculations and statistics, I cannot make an adequate statement.

Reviewer #2: Authors have addressed most of my previous comments, or at least give a reasonable reason the do in a different way.

7. PLOS authors have the option to publish the peer review history of their article (what does this mean?). If published, this will include your full peer review and any attached files.

Reviewer #1: No

Reviewer #2: No

---

## [Editor Report · Acceptance letter]

8 Mar 2021

PONE-D-20-37287R1 

Evaluating the state-of-the-art in mapping research spaces: a Brazilian case study 

Dear Dr. Murai:

I'm pleased to inform you that your manuscript has been deemed suitable for publication in PLOS ONE. Congratulations! Your manuscript is now with our production department. 

Kind regards, 

on behalf of

Professor Chi-Hua Chen 

Academic Editor

PLOS ONE